# Longitudinal Secretion of Paramyxovirus RNA in the Urine of Straw-Coloured Fruit Bats (*Eidolon helvum*)

**DOI:** 10.3390/v13081654

**Published:** 2021-08-20

**Authors:** Elli Rosa Jolma, Louise Gibson, Richard D. Suu-Ire, Grace Fleischer, Samuel Asumah, Sylvester Languon, Olivier Restif, James L. N. Wood, Andrew A. Cunningham

**Affiliations:** 1Institute of Zoology, Zoological Society of London, London NW1 4RY, UK; louise.gibson@ioz.ac.uk; 2Royal Veterinary College, Hatfield, Hertfordshire AL9 7TA, UK; 3School of Veterinary Medicine, College of Basic and Applied Sciences, University of Ghana, P.O. Box LG 25, Legon, Accra, Ghana; suuire@gmail.com (R.D.S.-I.); gfleischer16@gmail.com (G.F.); 4Wildlife Division of Forestry Commission, P.O. Box M 239, Accra, Ghana; ksasumah@gmail.com; 5West African Centre for Cell Biology of Infectious Pathogens (WACCBIP), Department of Biochemistry, Cell and Molecular Biology, University of Ghana, Legon, Accra 00233, Ghana; lansly19@gmail.com; 6Department of Veterinary Medicine, University of Cambridge, Cambridge CB3 0ES, UK; or226@cam.ac.uk (O.R.); jlnw2@cam.ac.uk (J.L.N.W.)

**Keywords:** chiroptera, *Pteropodidae*, *Henipavirus*, *Paramyxoviridae*, Rubulavirus, persistence

## Abstract

The straw-coloured fruit bat (*Eidolon helvum*) is widespread in sub-Saharan Africa and is widely hunted for bushmeat. It is known to harbour a range of paramyxoviruses, including rubuloviruses and henipaviruses, but the zoonotic potential of these is unknown. We previously found a diversity of paramyxoviruses within a small, captive colony of *E. helvum* after it had been closed to contact with other bats for 5 years. In this study, we used under-roost urine collection to further investigate the paramyxovirus diversity and ecology in this colony, which had been closed to the outside for 10 years at the time of sampling. By sampling urine weekly throughout an entire year, we investigated possible seasonal patterns of shedding of virus or viral RNA. Using a generic paramyxovirus *L*-gene PCR, we detected eight distinct paramyxovirus RNA sequences. Six distinct sequences were detected using a *Henipavirus*-specific PCR that targeted a different region of the *L*-gene. Sequence detection had a bi-annual pattern, with the greatest peak in July, although different RNA sequences appeared to have different shedding patterns. No significant associations were detected between sequence detection and birthing season, environmental temperature or humidity, and no signs of illness were detected in any of the bats in the colony during the period of sample collection.

## 1. Introduction

Bats (order *Chiroptera*) host a huge number and diversity of viruses and have been identified as the source of a range of recently emerged viruses of public health significance, including Hendra virus, Nipah virus, MERS coronavirus and probably SARS-CoV-2, the causative agent of COVID-19 [1,2]. Understanding the ecology of these viruses in their natural hosts, such as how they persist in bat populations, and possible risk factors for human infection is necessary for preventing zoonotic spill-over events [3,4]. Hendra and Nipah viruses (genus *Henipavirus*, family *Paramyxoviridae*) have caused human mortalities in Australia and Asia, respectively [5,6], and previous studies have detected closely related viruses in straw-coloured fruit bats (*Eidolon helvum*) in Africa, with antibodies against these viruses occurring in both the human and bat populations [7,8,9].

The transmission pathway for human exposure to bat viruses is not always clear. Hunting bats for bushmeat and living in an area undergoing deforestation are risk factors for seropositivity for bat paramyxoviruses in Africa [8]. Hendra virus infects people through horse intermediate hosts [10] and Nipah virus mainly through pigs or via the contamination of palm sap harvested for human consumption [11,12]. Most human cases of Hendra in Australia [13] and Nipah in Bangladesh follow seasonal patterns [14], and several studies looking into the shedding of paramyxoviruses in bat urine have detected seasonality [15,16]. This seasonality might be partially explained by the reproductive cycles of bats [16,17,18], although this effect has not been detected in all studies [19]. Straw-coloured fruit bats roost in large colonies of up to several million individuals and are migratory, making repeated sampling of the same individuals in the wild unlikely [20].

To enable more controlled studies of bat-virus dynamics, we established a research colony of straw-coloured fruit bats in 2009–2010, after which it was closed to contact with other bats [21]. Serologic studies of this colony have shown ~70% prevalence for *Henipavirus* antibodies in adult bats, the existence of maternal antibodies and a seasonal pattern in the seroconversion of the juveniles [21]. Modelling based on serologic data estimated that paramyxovirus persistence in the colony is most likely explained by a combination of reinfections and recurring latent infections [22]. In an earlier study, conducted in 2015, we detected nine different paramyxovirus RNA sequences using PCR analysis of urine collected under the bats roosting in the closed colony [23]. In this study, we aim to (1) determine if the paramyxovirus diversity in the closed colony has been maintained; (2) determine if the shedding of different paramyxoviruses in the colony is seasonal, intermittent or at a constant level throughout the year; and (3) evaluate possible risk factors for shedding.

## 2. Materials and Methods

This study was conducted in a closed captive breeding colony of straw-coloured fruit bats (*E. helvum*) in Accra, Ghana. The cage and the establishment of the colony were described by Baker et al. [21]. Briefly, the colony was established between July 2009 and January 2010 by capturing 77 wild bats from a natural roost approximately 6 km from the captive site. The captive colony is housed in a cage with a solid roof and double-walled sides to preclude contact with wild animals, including wild bats. At the beginning of 2019, the colony consisted of 154 individuals. During the study period, the bats were captured 3 times to be blood-sampled for a separate serologic study (on weeks 9, 28 and 47 of 2019), resulting in possible extra stress for them.

Under-roost urine samples were collected during 49 weekly and 12 extra sampling events, from February 2019 to January 2020, with two missing weeks in February and one in July. Three 1.5 metre × 3 metre tarpaulin sheets (Figure 1) were set under the bat roosting area at 4–7 pm, at the same time or soon after the bats were fed, and urine was collected 1–3 h later. During each sampling event, with the exception of one sampling (on the 7th of March in which 10 samples were collected), five 1–1.5 mL urine samples were collected from separate urine pools, resulting in a total of 310 pooled urine samples. Only visibly clean urine samples were collected, but faecal contamination could not be ruled out. Each urine pool was mixed in a 2 mL collection syringe before aliquoting 0.5 mL of each pooled sample into a separate vial containing 0.5 mL of the RNA preservation solution, RNA*later* (Invitrogen, Waltham, MA, USA), resulting in a 1:1 dilution of each sample. The remaining sample was stored in a plain Eppendorf tube for possible additional study. All samples were immediately transported in a frozen 1L Bio-Freeze container (Bio-Bottle, Auckland, New Zealand) to a −80 °C freezer for storage. At the time of each sample collection, the ambient temperature and humidity in the bat cage were recorded as were observations about the bats, such as the occurrence of pupping.

The molecular methods used in this study were almost identical to those of Gibson et al. [23], who described them in detail. The only difference was a different gel extraction kit used in this study. Briefly, RNA was extracted from 400 μL of each sample stored in RNA*later* using the MagMAX Viral RNA Isolation Kit (Applied Biosystems, Waltham, MA, USA) following the manufacturer’s protocol with the adjustment that carrier RNA was replaced with linear acrylamide. In order to remove any DNA in each sample, the RNA extracts were treated with TURBO DNA-free Kit (Ambion, Austin, TX, USA) following the manufacturer’s instructions. We ran two hemi-nested reverse transcription PCRs using primers that targeted two separate regions of the paramyxovirus L-gene: PAR-PCR that targeted a sequence shared among paramyxoviruses and RMH-PCR that targeted a sequence specific for *Respirovirus-Morbillivirus-Henipavirus*. The PCR method was modified from Tong et al. [24], who also describe the primers in detail. PCR products were run on 2% agarose gel and positive bands of appropriate size were gel extracted using the GeneJET Gel Extraction Kit (Thermo Scientific, Waltham, MA, USA) following the manufacturer’s protocol. Amplicons were submitted to a commercial laboratory (Eurofins Genomics, Ebersberg, Germany) for Sanger sequencing.

Sequences were initially aligned using MEGAX [25] and analysed using NCBI BLAST [26] to identify similarity with previously published sequences. Next, the sequences were aligned with the closest relatives derived from the BLAST search and reference sequences for other relevant paramyxoviruses from GenBank [26] using the MUSCLE [27] program in MEGAX [25]. Separate phylogenetic analyses were conducted for sequences obtained from each of the two PCRs with different primers (PAR and RMH). Maximum-likelihood phylogenetic trees were built using the GTR+I+G method [28] and bootstrapped 1000 times using MEGAX [25]. As the method of sampling might not have detected the total number of paramyxovirus sequences in the colony, and as paramyxovirus detection results were available from this colony from a previous study, we estimated the number of distinct paramyxovirus sequences by comparing the results from the 2015 study [23] and the current study with a capture-recapture method using a Chapman estimator [29,30]. For this, we used the package *recapr* version 0.4.3 [31] in R version 3.5.3 [32].

All statistical analyses were conducted in R version 3.5.3 [32]. Positive urine pools were plotted over time using the package *ggplot2* version 3.3.2 [33] and combined with the package *patchwork* version 1.1.0 [34]; sinusoidal seasonal patterns were analysed with the package *season* version 0.3.12 [35]. The five urine pools collected in each sampling event were non-independent; therefore, we defined a positive outcome for a sampling event as the detection of paramyxovirus RNA in at least one pool. The seasonality of positive observations was tested by fitting sinusoidal logistic regression models with 1, 2, 3 and 4 yearly cycles into the data [36]. To estimate risk factors, initial differences between positive and negative sampling events were compared using *t*-tests (temperature and humidity) and chi-square tests (birthing season, month and sampling events following a stressful event (capture for blood collection)). Associations between variables were explored using Pearson’s correlation (temperature vs. humidity) and *t*-tests (temperature vs. birthing season and humidity vs. birthing season). Associations between birthing season, temperature, humidity and the detection of viral RNA were analysed using multivariable logistic regression.

## 3. Results

### 3.1. Virus Diversity

We detected paramyxovirus RNA in 23.5% (73/310) of all urine pools using at least one of the PCR methods. In 12 pools, we detected viral RNA using both PCR methods, but none of the sequences from the different PCRs paired consistently together (Appendix A).

Eight distinct paramyxovirus sequences were detected using the general paramyxovirus PCR (PAR) (Figure 2). Two were within the genus Pararubulavirus (AZ_PAR_44, AZ_PAR_198), one in Orthorubulavirus (AZ_PAR_10B), and five were related to Henipavirus but too distant to be classified within that genus (“Henipa-like” viruses: AZ_PAR_3, AZ_PAR_117, AZ_PAR_162B, AZ_PAR_292, AZ_PAR_317B). Three PAR PCR sequences were novel with only moderate similarity with the closest relative in the NCBI database: AZ_PAR_10B (79.74% similarity), AZ_PAR_292 (75.33% similarity) and AZ_PAR_198 (70.22% similarity) (Appendix A).

Six distinct paramyxovirus sequences were detected using the Respirovirus-Morbillivirus-Henipavirus PCR (RMH) (Figure 3). All were related to Henipavirus but too distant to be classified within that genus (“Henipa-like” viruses: AZ_RMH_2, AZ_RMH_9, AZ_RMH_10A, AZ_RMH_146, AZ_RMH_162A, AZ_RMH_317A). None of the RMH PCR sequences were novel with 97.27–99.77% similarities with the closest relatives in the NCBI database (Appendix A).

The capture-recapture analysis was based on finding 2/6 of the RMH sequences and 1/3 of the PAR sequences initially identified in the captive colony in 2015 [23] in the current study (Figure 2 and Figure 3). This analysis resulted in an estimate of 36 (95% confidence intervals [CI]: 14–59) different paramyxovirus sequences in the research colony. When calculated for the different PCRs separately, the estimates for the number of different sequences were 17 (95% CI: 4–30) for PAR and 15 (95% CI: 6–25) for RMH.

### 3.2. Shedding Patterns

Paramyxovirus RNA was detected in at least one of the five urine pools in 60.7% (37/61) of all sampling events (Appendix A). Positive sampling events occurred throughout the year (Figure 4), but they were not evenly distributed over the months (χ2(11) =  20.3, *p* = 0.04). A significant seasonal pattern was detected with a biannual wave in positive samplings (*p* value for sine-wave = 0.003) with peaks in late July and late January (Figure 4). None of the other seasonal models (for 1, 3 or 4 cycles of viral RNA shedding) resulted in significant values. The two most commonly detected RNA sequences (AZ_PAR_10B and AZ_RMH_10A) were detected throughout the year, whereas most (8/14) other sequences were detected only on 1–3 occasions (Figure 5).

Newborn bats were observed from early March to September, peaking in July. There was no difference in the occurrence of positive sampling events between birthing and non-birthing seasons (χ2(1) = 0.3, *p* = 0.6), but one peak in the seasonal detection wave (Figure 4) coincided with the observed peak in pupping (July). Neither temperature nor humidity differed between negative and positive sampling events (*t*(59) = −0.2, *p* = 0.9 and *t*(59) = −1.2, *p* = 0.2, respectively). Temperature and humidity negatively correlated, albeit weakly (coefficient −0.3, *p* = 0.03), and their values did not differ between birthing and non-birthing seasons (*t*(59) = 1.7, *p* = 0.09 and *t*(59) = 0.8, *p* = 0.4, respectively). None of the variables tested had a significant effect on the occurrence of PCR-positive samples using multivariable logistic regression (Table 1). All pooled urine sampling events during the week following each bat capture for blood collection (7–10 days after the stressful event) resulted in paramyxovirus RNA detection (4/4), but the sample size was too small to allow meaningful comparison with RNA detections from other sampling events (33/57) (χ2(1) = 1.3, *p* = 0.3).

## 4. Discussion

We detected eight virus sequences using a generic paramyxovirus PCR and six virus sequences using a *Respirovirus-Morbillivirus-Henipavirus* specific PCR when we used these to analyse urine samples from our captive bat colony. Only three of the nine sequences from an earlier study (conducted in 2015; [23]) were detected in the current study. Three of the sequences detected in the current study were novel (only distant relatives in the NCBI database). We detected a seasonal pattern in paramyxovirus shedding in *Eidolon helvum* urine with the clearest peak in July, but we did not detect significant associations between virus shedding and birthing season, temperature, or humidity.

Some of the sequences obtained using the different PCR primers could have been different parts of the same virus, because the two sets of primers used in this study targeted different parts of the *L*-gene. This does not appear likely, however, as we did not detect any pairs of sequences consistently within the same samples. That said, there were two possible pairs in which all sequences were from “Henipa-like” viruses: AZ_PAR_117 was detected only 3 times and always with AZ_RMH_2, but AZ_RMH_2 was also detected alone in four samples and once with AZ_PAR_3; AZ_RMH_317 was detected only once and concurrently with AZ_PAR_317B, but AZ_PAR_317B was also detected alone twice.

Only AZ_PAR_162B, AZ_RMH_9 and AZ_RMH_162A had been detected previously in an earlier study (conducted in 2015; Gibson et al. submitted to this issue). In 2015, urine was collected from tarpaulin sheets that had been left under the colony overnight, whereas in the current study the samples were obtained within a few hours of starting under-roost urine collection. Even though no samples with visible faecal contamination were collected in either study, it is likely that levels of faecal contamination were higher in 2015 than in the current study. It is possible, therefore, that some of the 2015 sequences were shed predominantly in faeces or that the presence of faecal contamination could also have inhibited RNA detection [37] of viruses in the urine; thus the change in method could explain differences in the viral complement detected between the two studies. The more-extensive sampling spanning an entire year also could explain the greater number of virus sequences obtained in the current study. While cage design and biosecurity protocols would have greatly limited the likelihood of the transmission of PV infection from free-ranging bats to the captive colony [23], this was not impossible. If such an incursion had occurred, however, it seems highly unlikely that this would have happened for multiple PVs.

Based on the disparity in the number of viruses detected in 2015 and 2019, and with most (8/14) of the sequences having been detected only 1–3 times, the real number of different paramyxovirus sequences in the colony is likely to have been higher than 20 (distinct sequences from both studies combined). The estimate derived from the capture-recapture calculation (*n* = 36) is only a rough approximation for three reasons: the difference in sample collection methods between the years, the possibility that some viruses might have been lost from the captive bat population in the time between the studies and the possibility that we might have detected different parts of the same virus with the different PCRs. The range of paramyxoviruses detected in the current study agrees with previous evidence that bats are host to large numbers of paramyxoviruses and that these are typically within the genera *Henipavirus* and *Rubulavirus* or unclassified viruses close to the genus *Henipavirus,* whereas paramyxoviruses within the genera *Morbillivirus* and *Respirovirus* are more common in rodents [7].

The PCR methods employed in this study amplified two small fragments of the paramyxovirus *L*-gene, which is a conserved region coding for RNA polymerase [24]. As this region is not related to cell entry, it is not useful for estimating the degree of host specificity of these viruses [3]. Moreover, the detection of viral RNA does not mean that the bats shed infectious virus in their urine. To address these important questions, viral isolation and whole genome sequencing are required. Whole genome information will also help in estimating the actual number of paramyxoviruses in the colony.

We detected paramyxovirus RNA in bat urine throughout the year, but with uneven distribution of detections over time and among different sequences. Similar difference in shedding patterns across different viruses has been detected in Australian bat paramyxoviruses [38]. These findings imply that the risk for virus transmission from or among bats can vary over time and might have different risk factors for different paramyxoviruses. The small sample size and having observations from only one year limited the possibility of detecting significant risk factors for viral RNA shedding and the reliability of pattern detection. With these limitations in mind, we detected a significant two-peak wave pattern in positive observations. Stress is hypothesised to modify bat immunity and thus the amount of viruses that are shed into the environment [39]. Among potential stressors for wild bats are breeding cycles, migration, weather, food availability and human disturbance such as land use change [3,39].

We did not detect significant associations between shedding and temperature, humidity, or birthing season. Most free-ranging *E. helvum* bats are highly synchronous in giving birth, often prior to the annual peak in rainfall. In Uganda and south-western Nigeria this period is in February–March [40,41]; in Accra females in late pregnancy have been caught in March–April [20], before the local rainy season in May–June. Of note, a large roost with asynchronous breeding has been described in Kasanka, Zambia, with the hypothesis that this consists of bats that migrate from different areas with asynchronous breeding seasons [42]. A serological study conducted during the first two years after the establishment of the research colony showed that most seroconversions in juveniles, and either seroconversions or increases in the concentrations of henipavirus antibodies in females, took place in March 2011 and in January 2012 [21]. Those time periods were also observed to be when most females were in late pregnancy and showed a tight pupping synchronicity [21]. In that study, few adult males developed higher antibody levels or seroconverted, and any that did were not associated with breeding cycles [21]. During the current study, which took place in 2019/2020, the birthing season of the colony extended over 7 months with an observed peak in July, which is later and much less synchronous than that seen in the wild in West Africa. This peak coincides with the modelled peak in positive urine samples, during which all sampling events were positive for a 2-month period (Figure 4). As we do not have exact numbers of pups born in each month, however, this apparent association between birthing and virus shedding is subjective. The loss of pupping synchronicity in the captive bat population could be related to the year-round high availability of food due to provisioning, and this also could have affected the virus shedding pattern. Thus, the shedding patterns in this captive colony should be extrapolated to the wild with caution. No signs of illness were observed in the captive bats during the course of the current study, which is consistent with evidence that bats are able to be infected with various henipaviruses without detrimental effects [39].

Although not significant, the environmental variable we measured that was most closely, and positively, associated with virus shedding was humidity. This possible effect could be due to better henipavirus preservation in a moist environment [43]; signs of urine evaporation were sometimes observed during sampling events with low humidity. All of the urine collection events that took place within 7–10 days after a stressful event (i.e., the quarterly catch-up of the bat colony for blood serum collection) gave positive results for viral RNA. With only four such urine samplings throughout the study, however, any effect was not statistically significant. An earlier modelling study suggested that *E. helvum* bats shed paramyxoviruses for only short periods at a time [22] and a challenge study showed that Nipah and Hendra viruses are typically shed only for approximately 7 days post inoculation [44]. Studying the possible effect of handling stress on viral shedding, therefore, would require collecting samples on several days prior to and subsequent to a stressful event. This, and investigating the possible effect of food limitation on viral RNA shedding, are potential future directions for research.

The closed colony of *E. helvum* bats with only 77 founders has maintained infection with numerous paramyxoviruses for almost a decade. Due to using pooled urine, we were not able to explore virus shedding by individual bats because in any one urine pool, different viral sequences may have been shed by one or more bats. Further studies involving longitudinal urine sampling from individual bats might shed light on the issue. The rare detection of certain RNA sequences indicates that latent infections are likely, but it is also possible that urine shedding is not the main method of transmission for some of these viruses and thus, we might have detected only a small portion of virus shedding events. The evidence that *E. helvum* bats do not need large populations for paramyxovirus maintenance is inconsistent with the susceptible-infectious-resistant models of infection generally applied to paramyxoviruses but is rather more consistent with latent infections and/or waning immunity models. The results are consistent with previous serological evidence from this captive bat colony, and also from an isolated island population, both of which maintained high levels of seroprevalence in the absence of obvious routes for reinfection [21,45]. Models fitted to the longitudinal serologic data from this captive colony support a combination of resurgence of latent cases and the occurrence of reinfections [22]. This is consistent with the findings of a recent study, which showed that density-dependent transmission, viral recrudescence and the waning of acquired immunity are the main factors enabling Nipah virus persistence in populations of *Pteropus* spp. bats [46]. The persistence of paramyxoviruses in latently infected individuals occurs also in other mammals: in people, parainfluenza virus 5 can persist in the bone marrow [47], measles virus infection can recrudesce with subacute sclerosing panencephalitis years after the initial infection [48]; and in dogs, canine distemper virus can persist in the brain, causing old dog encephalitis [49]. However, the persistence of measles and canine distemper viruses is rare [48,49] and not normally associated with viral shedding, neither of which appears to be the case for paramyxoviruses in *E. helvum* based on the long-term maintenance of several distinct viral sequences shed from the captive colony investigated in this study. Future work, including whole genome sequencing of viruses, further longitudinal sampling of the colony and longitudinal sampling of individual bats, will help to identify mechanisms of infection persistence.

## 5. Conclusions

A small, closed colony of *Eidolon helvum* bats has maintained numerous paramyxoviruses for a decade without signs of illness. The general shedding of paramyxovirus RNA in their urine appears to follow a bi-annual pattern, and the shedding patterns appear to differ between paramyxoviruses. The persistence of several paramyxoviruses with infrequent RNA shedding adds to previous evidence that paramyxovirus maintenance in fruit bat populations is likely to be due to a combination of recrudescence of latent infections and reinfections through waning immunity.

## Figures and Tables

**Figure 1 viruses-13-01654-f001:**
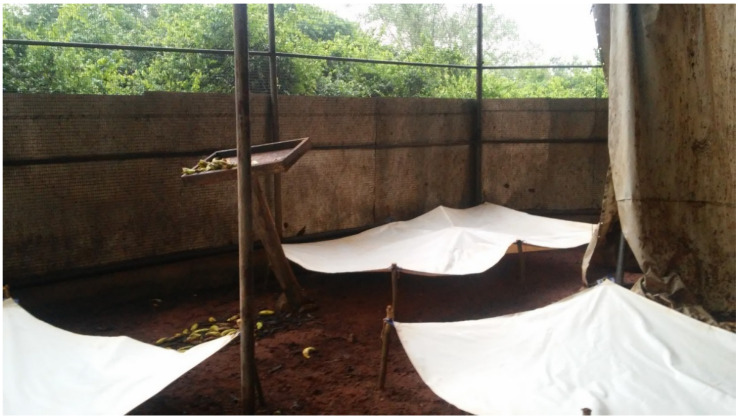
Three tarpaulin sheets set for under-roost urine sampling.

**Figure 2 viruses-13-01654-f002:**
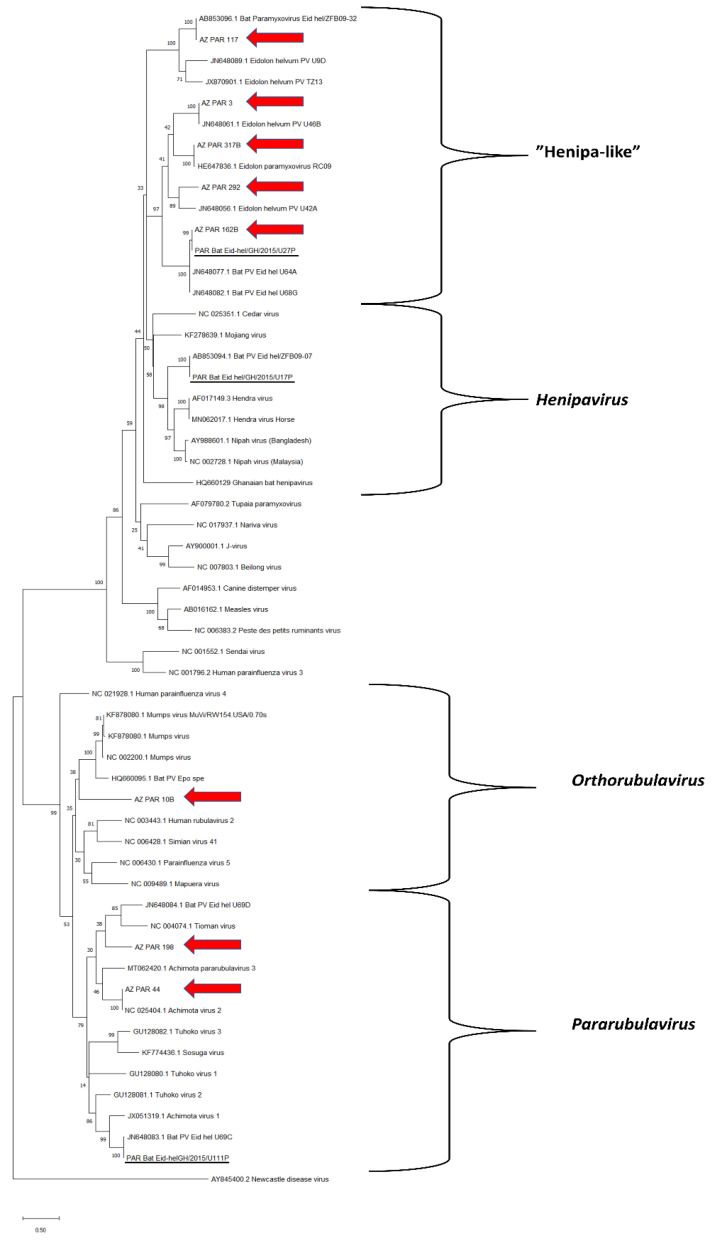
Maximum likelihood tree for sequences detected in the PCR using general paramyxovirus primers (PAR). The tree was rooted to the branch of Newcastle disease virus. The 2019 sequences from the research colony are highlighted with red arrows, and 2015 sequences reported by Gibson et al. [23] are underlined. All other sequences originate from the NCBI database. Bootstrap values for 1000 replicates are indicated as percentages, and nucleotide substitutions per site are to scale as indicated by the scale bar.

**Figure 3 viruses-13-01654-f003:**
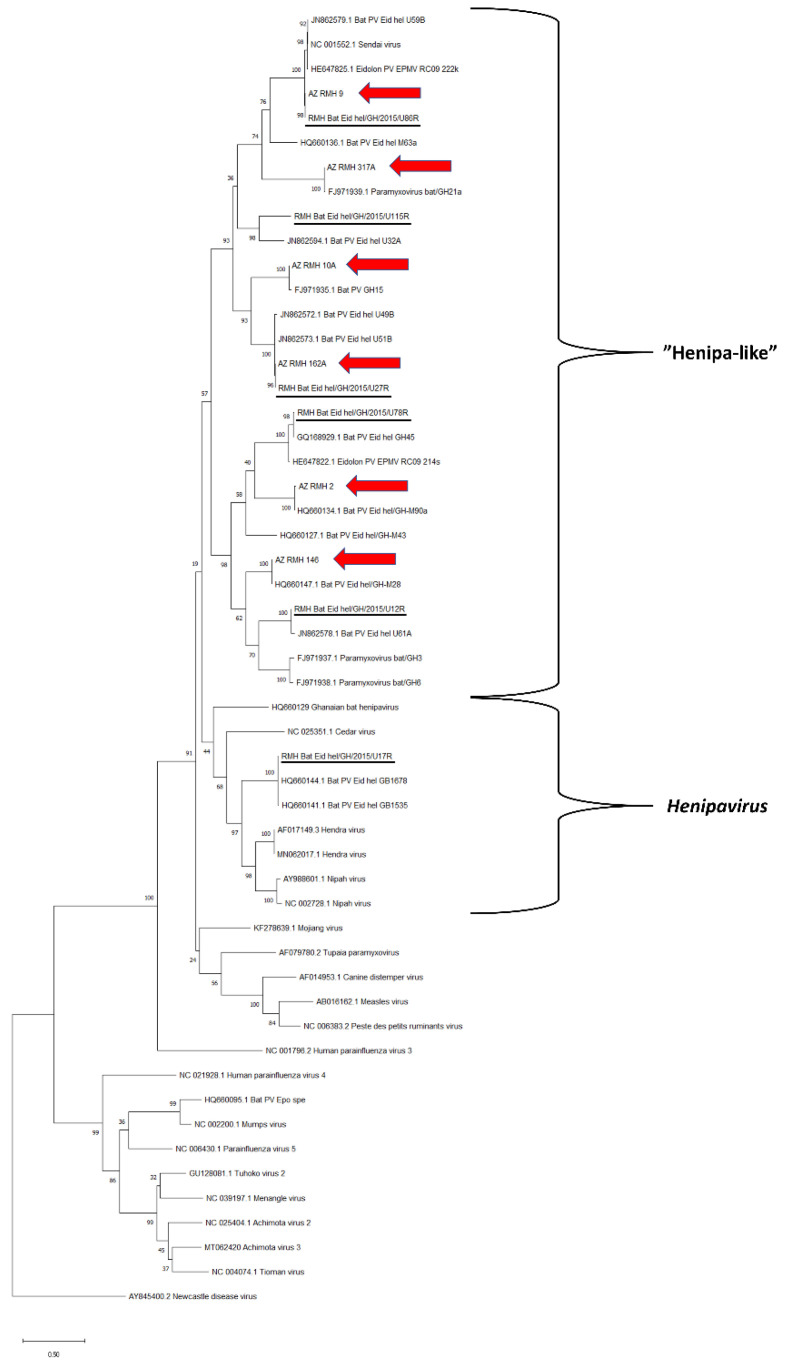
Maximum likelihood tree for sequences detected in the PCR using *Respirovirus-Morbillivirus-Henipavirus* specific primers (RMH). The tree was rooted to the branch of Newcastle disease virus. The 2019 sequences from the research colony are highlighted with red arrows, and 2015 sequences reported by Gibson et al. [23] are underlined. All other sequences originate from the NCBI database. Bootstrap values for 1000 replicates are indicated as percentages, and nucleotide substitutions per site are to scale as indicated by the scale bar.

**Figure 4 viruses-13-01654-f004:**
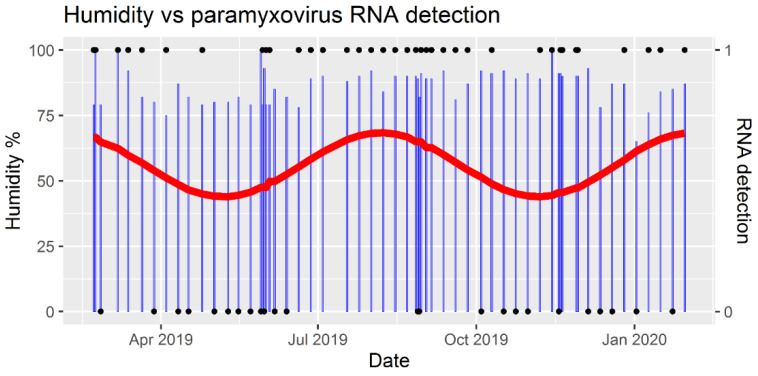
Detection of paramyxovirus RNA in pooled urine and air humidity over time. RNA detection (black dot at 1) means that at least one of the five samples collected in the sampling event tested positive for viral RNA; negative detection (black dot at 0) means that no paramyxovirus RNA was detected. Air humidity (blue bars) was recorded at the time of sample collection. The red line represents seasonal variation in RNA detections and was derived from a sinusoidal regression model with a biannual wave (*p*-value for the sine-wave = 0.003).

**Figure 5 viruses-13-01654-f005:**
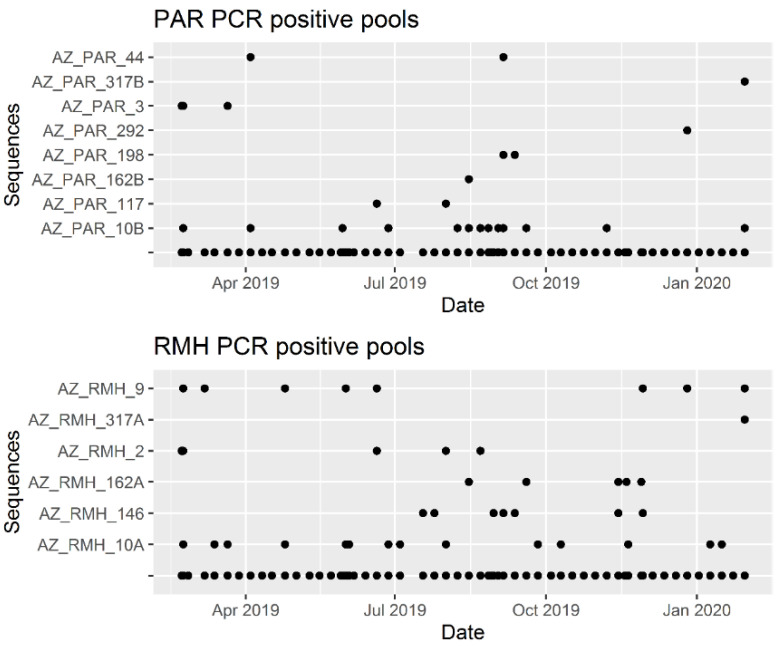
Detection of paramyxovirus RNA over time by the viral sequence. PAR = PCR with general paramyxovirus primers, RMH = PCR with *Respirovirus-Morbillivirus-Henipavirus* specific primers.

**Table 1 viruses-13-01654-t001:** Multivariable logistic regression predicting positive sampling event. Positive sampling event means that at least one of the five samples collected tested positive for viral RNA. Birthing season was the time when new births were detected in 2019 (March–September); temperature and humidity were recorded in the cage at the time of sampling. OR = odds ratio, CI = confidence intervals.

Variable	Crude OR (95% CI)	Adjusted OR (95% CI)	P (Wald’s Test)	P (Likelihood Ratio Test)
Birthing season	1.56 (0.55–4.46)	2.05 (0.63–6.69)	0.24	0.23
Humidity (%)	1.05 (0.97–1.14)	1.08 (0.97–1.19)	0.15	0.11
Temperature (°C)	1.03 (0.74–1.43)	1.19 (0.8–1.8)	0.39	0.37

## Data Availability

Data are contained within the article or supplementary material (Appendix A: The longitudinal sampling database). R code is available on request.

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
