# Peer review of "Longitudinal Secretion of Paramyxovirus RNA in the Urine of Straw-Coloured Fruit Bats (Eidolon helvum)"

_viruses, 2021, doi:10.3390/v13081654_

Round 1
Reviewer 1 Report
I think the manuscript reads well and have no comments to the content.
Author Response
I think the manuscript reads well and have no comments to the content.
We thank the reviewer for this observation.
Reviewer 2 Report
Jolma et al here longitudinally examine the shedding of Paramyxovirus RNA in the urine from a closed colony of Eidolon helvum bats over the course of a full year, with the goal of identifying evidence for persistent infections, spread of infections within the colony, and/or evidence for viral recrudescence. They also measure other variables that were hypothesized to affect rates of viral shedding and/or transmission within bat colonies, including birthing activity, temperature/seasonality, and humidity. Since humidity has the potential for a mechanical effect on viruses and RNA, it may also affect the researchers’ capactity to detect viral RNA in urine. They identify a number of different sequences shed by colony members with two peaks of detection throughout the year. One of these peaks corresponds to the “birthing” season of the bats, but multivariate analysis does not identify this as a significant correlation. The researchers note that the synchronicity of this colony’s birthing activity has broken down throughout captivity; this may impact efforts to correlate shedding and birthing activity. The researchers also note a spike in virus sequence detection in the week after the bats have been handled for blood sample collection (an acute stress), but this was of insufficient frequency to detect any significant correlation. Of interest, the researchers detect at least two sequences that are consistently shed without seasonality. Neither of these sequences were detected in a previous shorter study, intended to be co-published with this one, from samples collected roughly 5 years prior to these ones with similar methodology and from the same colony. This and their previous study lead the researchers to conclude that their small colony of roughly 115 bats harbours numerous persistent and periodically reactivating Paramyxoviruses, and has done so for nearly 10 years without overt disease.
Comments:
This is a thorough and interesting assessment of PV shedding in this bat colony, and an enjoyable read. Although the authors address some explanations for why the sequences identified here are largely disparate from those identified in 2015, I still find it remarkable that the overlap specifically does not include the two viruses that appear to be shed consistently in 2019-2020. This once again leads me to raise the possibility for introduction of viruses to the colony. See minor revision below.
The authors have maintained this colony for ten years; at this point, between banked and upcoming samples, the authors should have an avenue or planned research to experimentally assess whether these viruses are re-circulating or whether they are persistently infecting some (or all) animals in the colony (e.g. longitudinal analysis of RNA sequence from the serum of (a) known-positive bat(s)?). I would appreciate a mention of this plan, as well as information about whether multi-year studies are planned to provide more robust data about the seasonality of shedding in this colony. Similarly, it would be reassuring to know the researchers are intending to seek full-length versions of sequences from these samples.
Minor revisions:
Given the demonstrative image in Figure 1, I recommend the authors address the possibility of some outside introduction of PVs from wild bats by aerosolized respiratory and urinary secretions in their discussion. This would fit well in the discussion of previous calculations for number of hosts required to permit sustained transmission of an acute infection vs persistent infection.
I suggest the authors replace the term “breeding” (which often refers to mating) with either birthing (already used in the manuscript) or parturition, in both main text and supplementary table S3.
As in the co-submitted manuscript, the virus referred to here as “Ghana virus” should be re-named to “Ghanaian bat henipavirus” per ICTV recommendations in phyologenetic trees. The authors should also revisit the names of other viruses in the trees and simplify them (for example, “Sendai virus genomic RNA”, while the name given in NCBI, is unnecessarily complicated: “Sendai virus” alone would be clearer).
Phylogenetic trees should have scale bars. (Authors should revisit sister publication to verify the same.)
Author Response
This is a thorough and interesting assessment of PV shedding in this bat colony, and an enjoyable read. Although the authors address some explanations for why the sequences identified here are largely disparate from those identified in 2015, I still find it remarkable that the overlap specifically does not include the two viruses that appear to be shed consistently in 2019-2020. This once again leads me to raise the possibility for introduction of viruses to the colony. See minor revision below.
The authors have maintained this colony for ten years; at this point, between banked and upcoming samples, the authors should have an avenue or planned research to experimentally assess whether these viruses are re-circulating or whether they are persistently infecting some (or all) animals in the colony (e.g. longitudinal analysis of RNA sequence from the serum of (a) known-positive bat(s)?). I would appreciate a mention of this plan, as well as information about whether multi-year studies are planned to provide more robust data about the seasonality of shedding in this colony. Similarly, it would be reassuring to know the researchers are intending to seek full-length versions of sequences from these samples.
- Mention of planned future research has now been added to the Discussion, Lines 365-367
Minor revisions:
Given the demonstrative image in Figure 1, I recommend the authors address the possibility of some outside introduction of PVs from wild bats by aerosolized respiratory and urinary secretions in their discussion. This would fit well in the discussion of previous calculations for number of hosts required to permit sustained transmission of an acute infection vs persistent infection.
- We have now addressed this possibility in the Discussion, lines 260-264.
I suggest the authors replace the term “breeding” (which often refers to mating) with either birthing (already used in the manuscript) or parturition, in both main text and supplementary table S3.
- This revision has now been made throughout the manuscript (lines 137, 140, 141, 208, 213, 227, 238, 298, Table 1 and Supplementary table S3).
As in the co-submitted manuscript, the virus referred to here as “Ghana virus” should be re-named to “Ghanaian bat henipavirus” per ICTV recommendations in phyologenetic trees. The authors should also revisit the names of other viruses in the trees and simplify them (for example, “Sendai virus genomic RNA”, while the name given in NCBI, is unnecessarily complicated: “Sendai virus” alone would be clearer).
- The name has been changed to “Ghanaian bat henipavirus” in both trees and unnecessary extra information has been removed from other virus names.
Phylogenetic trees should have scale bars. (Authors should revisit sister publication to verify the same.)
- The trees in the original manuscripts were bootstrap consensus trees. Consensus trees show the topology with associated bootstrap values (the percentage of trees with that branch) and not evolutionary distances as branch topology and branch lengths are not the same in all 1000 trees. To address the reviewer’s concerns, we have changed the phylogenetic tress to maximum likelihood trees in order to show the evolutionary distance with branch lengths. The changes to the trees are reflected in the updated captions.